# High-throughput framework for genetic analyses of adverse drug reactions using electronic health records

Neil S. Zheng[1], Cosby A. Stone[2], Lan Jiang[3], Christian M. Shaffer[4], V. Eric Kerchberger[1,2], Cecilia P. Chung[3,4,5,6], QiPing Feng[5], Nancy J. Cox[6,7], C. Michael Stein[5,8], Dan M. Roden[1,5,8,9], Joshua C. Denny[1], Elizabeth J. Phillips[10], Wei-Qi Wei[1]*

1 Department of Biomedical Informatics, Vanderbilt University Medical Center, Nashville, Tennessee, United States of America, 2 Division of Allergy, Pulmonary and Critical Care Medicine, Department of Medicine, Vanderbilt University Medical Center, Nashville, Tennessee, United States of America, 3 Division of Rheumatology & Immunology, Department of Medicine, Vanderbilt University Medical Center, Nashville, Tennessee, United States of America, 4 Tennessee Valley Healthcare System—Nashville Campus, Nashville, Tennessee, United States of America, 5 Division of Clinical Pharmacology, Department of Medicine, Vanderbilt University Medical Center, Nashville, Tennessee, United States of America, 6 Vanderbilt Genetics Institute, Vanderbilt University Medical Center, Nashville, Tennessee, United States of America, 7 Division of Genetic Medicine, Department of Medicine, Vanderbilt University Medical Center, Nashville, Tennessee, United States of America, 8 Department of Pharmacology, Vanderbilt University, Nashville, Tennessee, United States of America, 9 Division of Cardiovascular Medicine, Department of Medicine, Vanderbilt University Medical Center, Nashville, Tennessee, United States of America, 10 Division of Infectious Diseases, Department of Medicine, Vanderbilt University Medical Center, Nashville, Tennessee, United States of America

* wei-qi.wei@vumc.org

**Data Availability Statement:** Data cannot be shared publicly because it includes confidential genetic and electronic health record data. Data is available from the VUMC Synthetic Derivative and

## Abstract

Understanding the contribution of genetic variation to drug response can improve the delivery of precision medicine. However, genome-wide association studies (GWAS) for drug response are uncommon and are often hindered by small sample sizes. We present a high-throughput framework to efficiently identify eligible patients for genetic studies of adverse drug reactions (ADRs) using "drug allergy" labels from electronic health records (EHRs). As a proof-of-concept, we conducted GWAS for ADRs to 14 common drug/drug groups with 81,739 individuals from Vanderbilt University Medical Center's BioVU DNA Biobank. We identified 7 genetic loci associated with ADRs at $P < 5 \times 10^{-8}$, including known genetic associations such as *CYP2D6* and *OPRM1* for CYP2D6-metabolized opioid ADR. Additional expression quantitative trait loci and phenome-wide association analyses added evidence to the observed associations. Our high-throughput framework is both scalable and portable, enabling impactful pharmacogenomic research to improve precision medicine.

## Author summary

Adverse drug reactions are a considerable burden on the healthcare system. Genetic studies can improve our understanding of the pathophysiological mechanisms of adverse drug

BioVU DNA BioBank (contact via victrbigdata@vumc.org) for researchers who meet the criteria for access to confidential data. Data has been deposited to the NCBI dbGaP under accession number phs002306.v1.p1 (https://www.ncbi.nlm.nih.gov/gap/study/status/40059).

**Funding:** The study was supported by National Institutes of Health, under grant numbers R01 HL133786 (W-QW), R35 GM131770 (CMS), P50 GM115305 (JCD, EJP, DMR), and R01 HG010863 (EJP). CPC was also funded by grants R01 AR073764 and the Veterans Health Administration Merit Award 1I01CX001741. The dataset used for the analyses described were obtained from Vanderbilt University Medical Center's resources, the Synthetic Derivative, which are supported by institutional funding and by the National Center for Advancing Translational Science grant 2UL1 TR000445-06 from NCATS/NIH. The funders had no role in study design, data collection, and analysis, decision to publish, or preparation of the manuscript.

**Competing interests:** The authors have declared that no competing interests exist.

reactions but have been hindered by small sample sizes. Drug responses are less often recorded than physiological traits and common diseases. Here, we present a high-throughput framework to efficiently identify eligible patients for genetic studies of adverse drug reactions from electronic health records. We validated our approach by conducting genome-wide association studies for adverse reactions to 14 common drug/drug groups with 81,739 individuals from Vanderbilt University Medical Centre's BioVU DNA Bio-bank, identifying 7 genetic loci associated with adverse drug reactions. Our high-throughput framework can enable impactful pharmacogenomic research to help develop clinical guidelines for the delivery of the right drug to the right person.

## Introduction

Genome-wide association studies (GWAS) have contributed substantially to precision medicine, providing critical insights into the physiological and pathophysiological mechanisms of human complex traits and diseases. [1,2] However, less than 10% of published GWAS have focused on drug response. [3] Adverse drug reactions (ADRs) are a considerable burden on patients and healthcare systems as a major source of hospitalization, morbidity, and mortality. [4–7] The lack of such pharmacogenomics GWASs on ADRs hinders our ability to deliver the right drug to the right person. [3,6–9]

A significant challenge for pharmacogenomics discovery is small sample size. [3,10] Drug response phenotypes, such as ADRs, are less often recorded than physiological traits and common diseases. [3,7,10] Traditional studies that recruit patient cohorts remain cumbersome and costly, and usually result in limited statistical power to detect genetic predictors with small effect sizes. [3,7,10] Biobanks that are linked to electronic health records (EHRs) can generate large datasets for efficient *discovery and replication GWASs*. [7,10,11] However, defining drug response using EHR data (i.e., pharmacological phenotyping), remains difficult. Unlike disease phenotypes, which can be represented with diagnostic codes, drug response information is often embedded in clinical notes, [11,12] complicating the development and implementation of uniform methods to extract drug response phenotypes. [11,13]

In this study, we investigated the feasibility of using the allergy section in EHRs to conduct high-throughput GWAS of reported ADRs. In routine practice, healthcare providers often use "drug allergy" labels in an allergy section to document a patient's intolerance or allergy to a drug as reported by the patient or observed by a healthcare provider. [14,15] Despite being called an "allergy" section, the documented information most clearly satisfies the definition for ADR, which includes any noxious, unintended or undesired effect of a drug experienced at normal therapeutic doses. [7,16] The ADR information in this section is meant to be reconciled with every patient encounter to capture new information. The allergy section is semi-structured (i.e., some structure but does not adhere to any rigorous format), which allows for easy retrieval of adverse reaction information without *sophisticated natural language processing*, enabling high-throughput analysis when linked to genetic data.

We hypothesized that drug allergy labels from the allergy sections in EHRs can be leveraged for efficient identification of reported ADRs. We developed and applied a high-throughput approach for identifying ADRs from allergy sections in EHRs from Vanderbilt University Medical Center's (VUMC) Synthetic Derivative. Then using VUMC's BioVU DNA Biobank, [17] we conducted GWAS on 14 drug (drug class) ADRs in a subset of 67,323 individuals with self-reported European ancestry (EA), followed by trans-ethnic validation in 14,416 individuals with self-reported African ancestry (AA). Additional expression quantitative trait loci

(eQTL) analyses and phenome-wide association analyses (PheWAS) were performed on the lead variants. [18,19]

## Results

### Identifying adverse drug reactions in electronic health records

A summary of selected EHR characteristics for all individuals with available EHRs at VUMC and the selected BioVU individuals is shown in Table 1. The BioVU cohort had mean EHR length of 10.6 years, which was more than double the length of the mean EHR length for all VUMC individuals (4.4 years). Additionally, a greater proportion of BioVU individuals (95.3%) had information documented in their allergy section compared to all VUMC individuals (62.4%). Similarly, a greater proportion of BioVU individuals (63.0%) had at least one reported ADR compared to all VUMC individuals (28.6%). While the proportion of individuals that have information documented in their allergy section is similar between EAs and AAs, we observed that the proportion of individuals with reported ADRs was greater among EAs compared to AAs for all VUMC individuals and the BioVU cohort.

The most frequently documented ADRs were to penicillins (17.4%), sulfa drugs (11.6%), and codeine (9.1%). Cases and controls for GWAS of 14 adverse drug or drug group reactions are shown in Table 2. We selected the top 10 most frequent drugs or drug classes reported in the allergy sections: penicillins, sulfa drugs, codeine, morphine, aspirin, lisinopril, levofloxacin, erythromycin, meperidine, and cephalexin. The top 10 most frequently reported drugs in the allergy sections were the same for EAs and AAs with differences in ordering. Additionally, we observed that ADRs to statins as a class of drugs were reported frequently. Therefore, we identified ADRs to any statin for a grouped analysis since the class of drugs shares a similar metabolic pathway and further broke down ADRs into atorvastatin only or simvastatin only. Likewise, we selected CYP2D6-metabolized opioid prodrugs, including codeine, hydrocodone, oxycodone, and tramadol, as a grouped analysis. [20] Types of adverse drug reactions for the 14 selected drug or drug groups are summarized in S1 Table. The type of reaction is not always documented in the allergy section and the percent missing ranges from 24.4% to 58.0%.

### Genome-wide analysis

The genetic analyses for EAs identified genome-wide significant signals ($P < 5 \times 10^{-8}$) for 7 of the 14 adverse drug reactions. The lead variant for each signal is shown in Table 3, and additional correlated variants are reported in S2 Table. The trans-ethnic validation of the identified signals for EAs in the AA cohort yielded no significant findings (S3 Table). Genome-wide

**Table 1. Summary of selected EHR characteristics for all VUMC individuals and the BioVU individuals selected for genetic analyses, stratified by self-reported ancestry.**

| Cohort | N | EHR length (years), Mean ± SD | Have allergy section* | At least one ADR* |
|---|---|---|---|---|
| All individuals | 3,169,625 | 4.4 ± 6.0 | 1,979,220 (62.4) | 905,301 (28.6%) |
| European ancestry | 1,957,846 | 5.3 ± 6.3 | 1,376,127 (70.2) | 679,141 (34.7%) |
| African ancestry | 310,864 | 6.3 ± 7.0 | 214,149 (68.8) | 77,223 (24.8%) |
| Other | 900,915 | 1.6 ± 3.4 | 388,944 (43.2) | 148,937 (16.5%) |
| BioVU individuals | 81,739 | 10.6 ± 7.3 | 77,907 (95.3) | 51,534 (63.0) |
| European ancestry | 67,323 | 10.7 ± 7.2 | 64,166 (95.3) | 44,407 (66.0) |
| African ancestry | 14,416 | 10.0 ± 7.8 | 13,714 (95.3) | 7,127 (48.4) |

* Reporting count and row percentages for the respective cohort

**Table 2. Case and control counts for adverse drug reactions to 14 selected drugs or drug groups, stratified by self-reported ancestry.**

| Drugs/Drug Groups | European ancestry (N = 67,323) | | African ancestry (N = 14,416) | |
|---|---|---|---|---|
| | Cases (%)[a] | Controls | Cases (%) | Controls |
| Penicillin | 12294 (18.3) | 38284 | 1894 (13.1) | 9539 |
| Sulfa | 8492 (12.6) | 46642 | 964 (6.7) | 11085 |
| Codeine | 6706 (10.0) | 24579 | 706 (4.9) | 7330 |
| Morphine | 3646 (5.4) | 38181 | 450 (3.1) | 9515 |
| Aspirin | 1800 (2.7) | 47351 | 401 (2.8) | 10264 |
| Lisinopril | 1591 (2.4) | 34096 | 439 (3.0) | 8959 |
| Levofloxacin | 1737 (2.6) | 35888 | 136 (0.9) | 8560 |
| Erythromycin | 1607 (2.4) | 23400 | 138 (1.0) | 7310 |
| Meperidine | 1499 (2.2) | 27261 | 112 (0.8) | 7590 |
| Cephalexin | 1460 (2.2) | 30479 | 124 (0.9) | 7995 |
| Any statin | 2927 (4.3) | 42551 | 258 (1.8) | 9897 |
| Atorvastatin | 1325 (2.0) | 31048 | 86 (0.6) | 8234 |
| Simvastatin | 1020 (1.5) | 31394 | 111 (0.8) | 8053 |
| CYP2D6-metabolized opioids[b] | 10264 (15.2) | 48445 | 1343 (9.3) | 11288 |

[a] Reporting count and percentage of self-reported ancestry population identified with ADR

[b] CYP2D6-metabolized opioids include codeine, hydrocodone, oxycodone, and tramadol

analyses in AA individuals were excluded in our primary analysis due to the potential for unstable point estimates and inflated false discovery rates from limited sample size. Nonetheless, significant ADR-genetic associations in AAs may be informative for future studies and have been included S4 Table.

The opioids shown in Fig 1 are prodrugs metabolized to a morphine or morphine-like active metabolites by CYP2D6. We identified a strong genome-wide significant association signal near the *CYP2D6* gene for codeine and CYP2D6-metabolized opioid ADRs (Fig 1). Near the *CYP2D6* locus, the minor allele of the variant rs9620007 (G) was associated with reduced risk of codeine ADRs (Odds ratio [OR] = 0.84; 95% confidence interval [CI] = 0.79 to 0.89) and CYP2D6-metabolized opioid ADRs (OR = 0.86; 95% CI = 0.82 to 0.90). Additionally, the

**Table 3. Lead variant per signal associated with adverse drug reactions for European ancestry patients.**

| Adverse Drug Reaction | Variant | Mapped Gene | Consequence | Allele[a] | EAF | $R^2$ | OR (95% CI)[b] | P |
|---|---|---|---|---|---|---|---|---|
| Aspirin | rs115346678 | *SSBP2, ATG10* | intergenic | G/A | 0.01 | 0.98 | 2.03 (1.79 to 2.28) | $1.40 \times 10^{-8}$ |
| Cephalexin | rs34545984 | *LOC105376453, OTUD1* | intergenic | G/T | 0.01 | 0.50 | 2.03 (1.79 to 2.28) | $1.23 \times 10^{-8}$ |
| Codeine | rs9620007 | *WBP2NL (CYP2D6)* | intronic | C/G | 0.30 | 0.98 | 0.84 (0.79 to 0.89) | $1.24 \times 10^{-13}$ |
| CYP2D6-metabolized opioids | rs62436463 | *OPRM1* | intronic | C/T | 0.10 | 0.94 | 0.84 (0.79 to 0.90) | $5.43 \times 10^{-10}$ |
| | rs739296 | *SEPTIN3 (CYP2D6)* | intronic | G/A | 0.30 | 0.99 | 0.86 (0.83 to 0.90) | $1.08 \times 10^{-16}$ |
| Meperidine | rs11049274 | *PTHLH, LOC729291* | intergenic | G/A | 0.08 | 0.99 | 1.42 (1.30 to 1.54) | $2.09 \times 10^{-8}$ |
| | rs113100019 | *FIP1L1* | intronic | T/G | 0.01 | 0.82 | 2.10 (1.84 to 2.36) | $2.26 \times 10^{-8}$ |
| | rs185462714 | *SERINC5* | intronic | A/G | 0.01 | 0.82 | 2.09 (1.83 to 2.35) | $3.37 \times 10^{-8}$ |
| Penicillin | rs115200108 | *HLA-B, MICA-AS1* | intergenic | C/A | 0.02 | 0.99 | 1.30 (1.21 to 1.39) | $4.23 \times 10^{-9}$ |
| Simvastatin | rs76103438 | *DIPK2A, LNCSRLR* | intergenic | T/A | 0.03 | 0.90 | 1.88 (1.65 to 2.09) | $2.56 \times 10^{-8}$ |

EAF = Effect allele frequency; $R^2$ = imputation quality

[a] Alleles are listed as reference/effect and are reported in the forward strand.

[b] OR and 95% CIs were derived from logistic regression models adjusted for sex, age, length of electronic health records (years), and first 10 principal components.

## A Manhattan plots

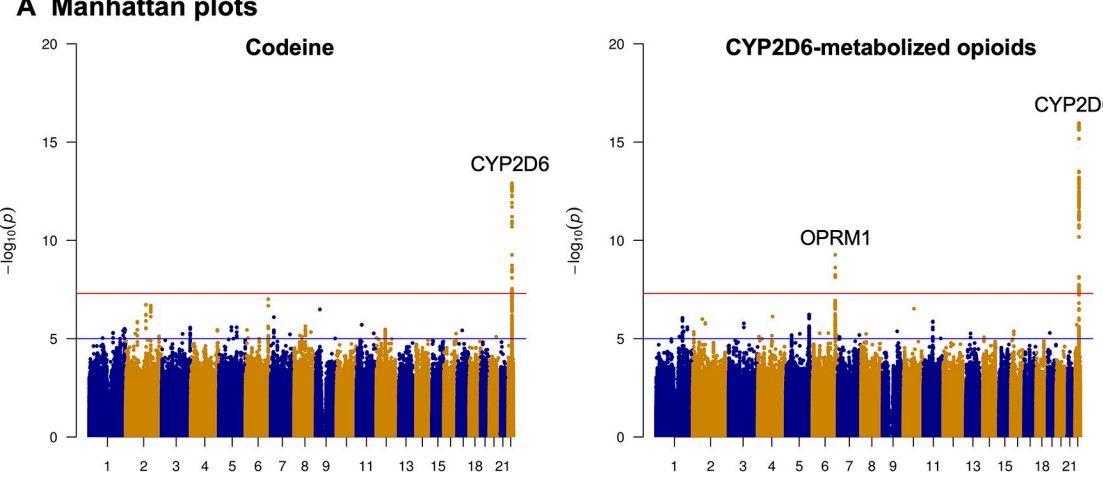

## B *CYP2D6* locus for CYP2D6-metabolized opioid adverse drug

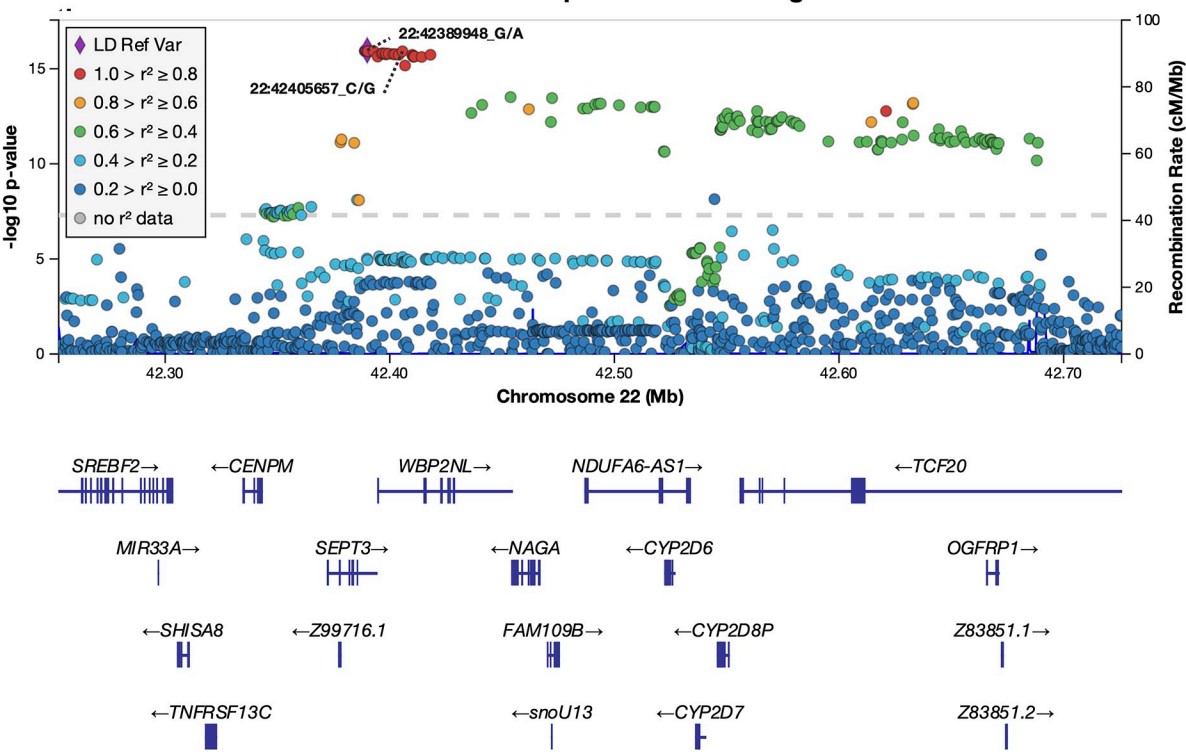

**Fig 1. A)** Manhattan plots of genome-wide association studies (GWAS) for codeine (left) and CYP2D6-metabolized opioid (right) adverse drug reactions (ADRs). Red lines on Manhattan plots show the genome-wide significance level ($P < 5.0 \times 10^{-8}$). **B)** *CYP2D6* locus for CYP2D6-metabolized opioid ADRs. SNPs are colored according to their linkage disequilibrium (LD, based on 1000 Genome phase3 EUR reference panel) with the lead variant rs739296 (22:42389948), which is marked with a purple diamond. The lead variant rs9620007 (22:42405657) for codeine ADRs is also labeled. Dotted gray line shows the genome-wide significance level ($P < 5.0 \times 10^{-8}$).

nearby variant rs739296 (A) was associated with reduced risk of CYP2D6-metabolized opioid ADRs (OR = 0.86; 95% CI = 0.83 to 0.90). The rs739296 (A) variant was also associated with reduced risk of specifically nausea/vomiting reactions to CYP2D6-metabolized opioids (OR = 0.80; 95% CI = 0.74 to 0.86). We found a significant association for *OPRM1* and

CYP2D6-metabolized opioid ADRs, where individuals carrying the minor allele of the lead variant rs62436463 (T) were less likely to have a reported ADR (OR = 0.84; 95% CI = 0.79 to 0.90). Notably, the minor allele of the exonic variant rs1799971 (G) in *OPRM1*, which is in high LD with the lead variant rs62436463, was also associated with reduced risk of CYP2D6-metabolized opioid ADRs (OR = 0.86; 95% CI = 0.82 to 0.91).

For meperidine ADRs, the analysis revealed a genome-wide significant association signal upstream of *PTHLH* and two significantly associated variants in *FIPL1* and *SERINC5* (Fig 2A). Additionally, we identified a genome-wide significant signal in the major histocompatibility

## A *PTHLH*-risk locus for meperidine adverse drug reaction

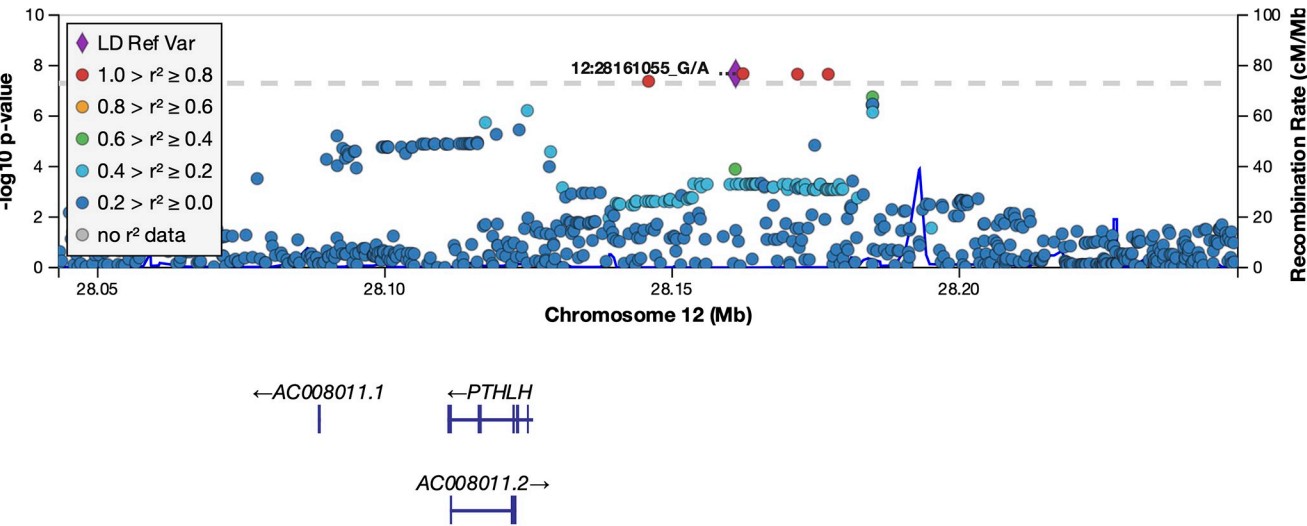

## B *HLA/MICA*-risk locus for penicillin adverse drug reaction

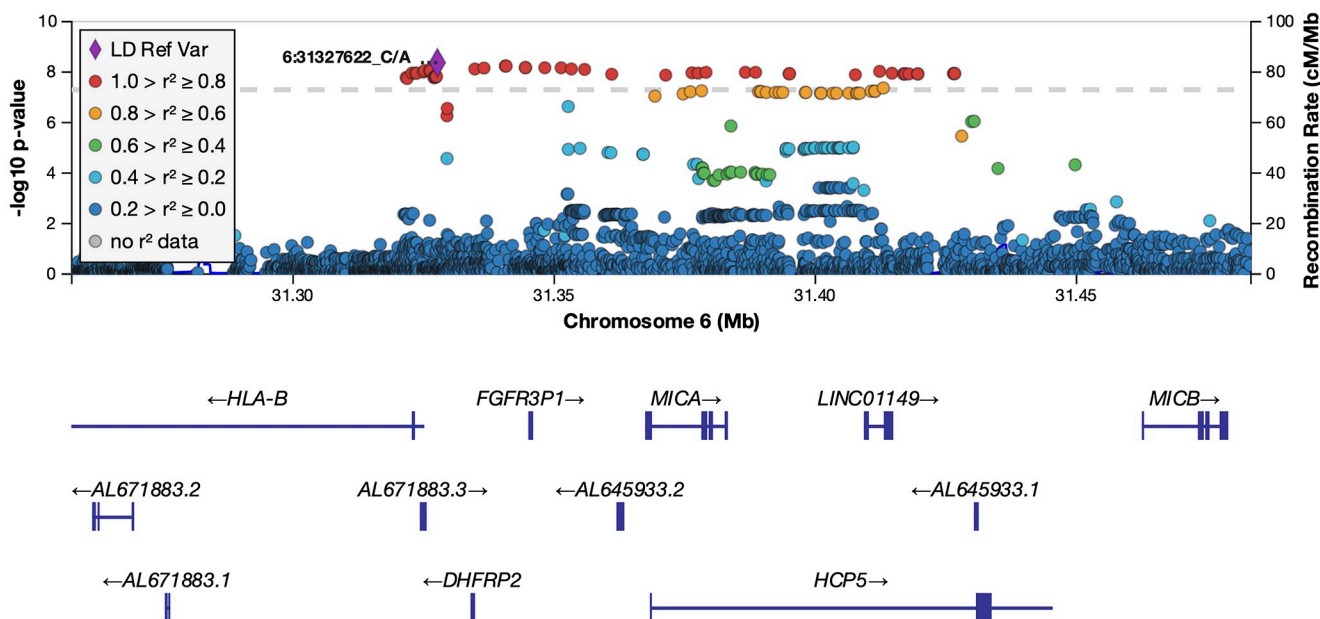

**Fig 2.** Risk loci for meperidine (**a**) and penicillin (**b**) adverse drug reactions (ADRs). SNPs are colored according to their linkage disequilibrium (LD, based on 1000 Genome phase3 EUR reference panel) with the lead variants rs11049274 (12:28161055) for meperidine ADRs and rs115200108 (6:31327622) for penicillin ADRs, which are marked with a purple diamond. Dotted gray line shows the genome-wide significance level ($P < 5.0 \times 10^{-8}$).

complex (MHC) region for penicillin ADR (Fig 2B). The minor allele of the lead variant rs115200108, which is located between *HLA-B* and *MICA*, was significantly associated with increased risk of penicillin ADRs (OR = 1.30, 95% CI = 1.21 to 1.39).

We also identified three low-frequency variants (minor allele frequency [EAF] < 0.05) that were strongly associated with ADRs to aspirin (rs115346678; OR = 2.03; 95% CI = 1.79 to 2.28), cephalexin (rs34545984; OR = 2.03; 95% CI = 1.79 to 2.28), and simvastatin (rs76103438; OR = 1.88; 95% CI = 1.65 to 2.09).

## Expression quantitative trait loci analyses

Using data from the Genotype-Tissue Expression (GTEx) project, we evaluated the correlation of the lead variants for the genetic loci identified by the GWAS and expression levels of putative target genes. For CYP2D6-metabolized opioid ADRs, the A allele of lead variant rs739296 in the *CYP2D6* locus was most significantly associated with decreased *WBP2NL* expression in adipose tissue (normalized effect size [NES] = -0.33; $P = 1.9 \times 10^{-11}$) and increased *CYP2D6* expression in brain tissue (NES = 0.55; $P = 5.3 \times 10^{-11}$). The T allele of the lead variant rs62436463 and the G allele of the exonic variant rs1799971 in *OPRM1* were both associated with higher *OPRM1* expression in the cerebellum with NES of 0.70 ($P = 9.5 \times 10^{-8}$) and 0.63 ($P = 1.4 \times 10^{-7}$), respectively.

The A allele of the lead variant rs11049274 in *PTHLH* for meperidine ADRs was significantly associated with increased *PTHLH* expression in muscoskeletal tissue (NES = 0.28; $P = 6.5 \times 10^{-5}$). Additionally, the A allele of rs115200108 for penicillin ADRs was most significantly associated with higher *MIR6891* expression in adipose tissue (NES = 1.3; $P = 2.0 \times 10^{-13}$) and reduced *MICA* expression in whole blood tissue (NES = -0.72; $P = 2.0 \times 10^{-13}$).

## Phenome-wide analyses

To compare our framework with the ability of diagnosis codes to identify ADRs, we performed PheWAS of the lead variants from the identified genetic loci (*CYP2D6, OPRM1, PTHLH, HLA/MICA*) (S1 Fig). The lead variant rs115200108 in the *HLA/MICA* risk-locus was associated with increased risk of 'Poisoning by antibiotic' with an (OR = 2.37; 95% CI = 1.90 to 2.84; $P = 3.0 \times 10^{-4}$) but did not reach phenome-wide significance ($P < 5.0 \times 10^{-5}$).

## Discussion

In this study, we present a high-throughput and scalable approach to conduct large-scale, genome-wide analyses for adverse drug reactions. Our framework can be adapted or shared between institutions, helping facilitate collaboration between sites. Utilizing EHRs allowed us to study ADRs in individuals with diverse clinical and ethnic backgrounds under the conditions of routine clinical care. As shown in this study, what and how physicians choose to document clinical observations or patients' self-reported details as drug allergies in the EHR may provide useful information. In addition, our results demonstrated the potential of utilizing EHRs and our framework to efficiently generate pharmacogenomic findings, which can provide insights for optimizing drug therapy with maximal efficacy and minimal adverse effects.

We found that 28.6% of individuals at VUMC had at least one drug listed in the allergy section of their EHRs. This is consistent with other studies have reported between 20 to 35 percent of their populations have at least one drug allergy label in their EHRs. [14,21] The genotyped BioVU cohort is a patient cohort (i.e., receives more frequent medical care than general population) and has more dense EHR data, which may explain the higher proportion of the BioVU cohort (66.0%) that reported at least one ADR. We also observed a lower proportion of reported ADRs among AAs than EAs, which is consistent with a previous report. [14]

As noted by the previous study, the difference in the reported ADRs between AAs and EAs may reflect a documentation bias that has been reported in other clinical domains. [14]

Using our ADR case-control definitions, analyses identified genetic loci for 7 of the 14 selected drug/drug group allergies. We found that variants in two well-known genetic loci, *CYP2D6* and *OPRM1*, were associated with reduced risk of CYP2D6-metabolized opioid ADRs. The analysis of eQTL data from the GTEx project showed that variants in the *CYP2D6* locus and in *OPRM1* were associated with elevated expression of these genes in the brain. [22] Previous studies have implicated both of these genetic loci in opioid response and metabolism. [23–25] Notably, an independent report on variants associated with reduced risk of opioid-induced vomiting in a 23andMe cohort supported our findings that the minor alleles of rs9620007 near *CYP2D6* and rs1799971 in *OPRM1* were associated with reduced risk of CYP2D6-metabolized opioid ADRs. [26] Furthermore, our analysis of CYP2D6-metabolized opioid related nausea or vomiting also identified the same loci near *CYP2D6* as associated with reduced risk. However, CYP2D6 metabolic activity also varies greatly depending on a copy number variation, [23] which was not available for this study. Therefore, further work is needed to better understand the contributions of genetic variations to *CYP2D6*-metabolized opioid ADRs. Additionally, studies have reported that patients who carried the G allele of rs1799971 in *OPRM1* required higher doses of opioid for pain relief. [27,28] It is possible that patients carrying the minor allele for the significant variants in *OPRM1* experienced reduced opioid effectiveness, which may affect their opioid sensitivity and risk of adverse reaction depending on the opioid dosage.

We also identified *HLA-MICA* as a risk-locus for penicillin ADR, which is supported with a recent large-scale genetic analysis for penicillin allergy including data from UK Biobank, Estonian Biobank and BioVU. The previous study also showed a strong association between penicillin allergy label and the *HLA-MICA* region with a different lead variant. [29] The eQTL analysis showed that the minor allele of the lead variant rs115200108 in the *HLA-MICA* risk-locus for penicillin ADR was associated with reduced *MICA* expression in whole blood tissue. The PheWAS results found that the minor allele of rs115200108 was highly associated with increased risk of 'Poisoning by antibiotic,' but did not reach phenome-wide significance. This finding suggests that our approach to identifying ADRs not only offers ADR phenotypes that are not covered by diagnosis codes but may also provide more power for genetic analyses than using diagnostic codes alone.

There have been no previous studies regarding the associations between *PTHLH* and meperidine allergy. In our eQTL analysis, we found that the lead variant in the *PTHLH* risk-locus for meperidine allergy was associated with increased *PTHLH* expression in muscoskeletal tissue. However, further investigation is needed to confirm this finding. Trans-ethnic validation among individuals with self-reported African ancestry did not replicate any associations of genome-wide significance, but this analysis may have been limited by smaller sample size. Additionally, we performed genetic imputation with reference panels from the Haplotype Reference Consortium, which were developed with individuals from predominantly European ancestry and therefore may not be adequate for individuals with self-reported African ancestry. [30] Likewise, genome-wide analyses in the African ancestry cohort were also limited by small sample sizes and predominantly European ancestry genetic reference panels. Further improvements in ADR documentation and genetic reference panels as well as the continued growth of EHR data may help us determine the generalizability of these findings in diverse populations. Due to the high-throughput nature of our framework, it should be easy to adapt to other large multi-ancestry EHR-based biobanks for future analyses.

There are several additional limitations to this study and approach. Drug allergy labels in the allergy section are entered into the EHRs by healthcare providers, but this information is

often self-reported or subject to interpretation bias by the individual receiving the information and entering the data, introducing potential documentation or selection bias. For instance, patients who communicate with their healthcare provider more frequently, whether due to their specific conditions or due to socio-behavioral factors, may be more likely to report their adverse drug reactions. A better understanding of the factors that affect the likelihood of receiving a drug allergy label may improve our ability to utilize EHRs to study ADRs. Additionally, it is likely there were some misclassification errors in the controls. Controls who were exposed to the drug and experienced an adverse reaction may not have reported the reaction to their clinician to be documented. Similarly, controls who were never exposed to the drug and only had the "no known drug allergy" label may experience an adverse reaction when exposed to the drug. However, misclassifications of cases as controls most likely biases the results to null and leads to an underestimation of the true contribution of genetic variation to ADRs.

While a drug allergy labels in the allergy section is consistent with a previous adverse drug reaction to the drug, more detailed questioning often reveals that a true allergy is less certain. [15,31] For instance the vast majority of patients who are labeled as having a penicillin allergy were typically labeled much earlier in childhood. [32–34] Studies in allergy practice show that >95% of these individuals that undergo validated skin testing and challenge will tolerate penicillin, in part due to waning of this allergic response over time. [31] Therefore, our analysis did not consider the possibility of patients having lost their allergic tendency and being delabeled for a drug allergy, and our results should be explained as 'ever or never' reported an adverse reaction to a drug. Indeed, it is more challenging to capture specific details in the EHR when identifying individuals who ever had a penicillin allergy label, rather than those who currently have a penicillin allergy.

We also observed that clinicians often do not enter information in the allergy section in a standardized manner, especially in older EHRs. Drug allergies and drug intolerances are frequently documented together in the allergy section without clear distinguishers. In addition, allergy section entries often omit details such as severity, type of reaction (e.g., anaphylaxis vs. rash), specific dose, and time of administration, limiting nuanced analyses. Although the CYP2D6-metabolized opioid related nausea/vomiting findings demonstrate that our framework can extract more detailed ADR phenotypes, the frequency of missing reaction information hinders a high-throughput analyses of specific adverse effects. Thus, the high-throughput nature of our framework means that our genetic analyses were likely driven by the milder, more frequent reactions (e.g., rash from penicillin) rather than rarer phenotypes like Stevens-Johnson syndrome. Nonetheless, genetic variants identified with our framework need further follow-up to better understand the potential risks of a medication for a patient. For instance, labeling a patient to be broadly 'at risk' for an ADR may cause the patient to be given suboptimal therapy even if the reaction may be a common, expected side effect.

These observation highlights the need to emphasize efforts to capture more accurate and relevant drug response information. Our framework will yield better outcomes as newer EHR systems introduce more explicit semantic meaning (e.g., allergy vs. intolerance), structured inputs and questionnaires (e.g., drop-down menus or checkboxes), [15] and increased quantity of quality data to the allergy section. Although these improvements *require time* and planning, it is encouraging that our current study in the context of these limitations can successfully identify several known genetic associations for ADRs.

In summary, our results demonstrate the utility and efficacy of a high-throughput framework to identifying ADRs and eligible individuals from EHRs for large-scale studies. Our approach is scalable and portable and can help accelerate the pace of impactful pharmacogenomic research for advancing precision medicine.

## Methods

### Ethics statement

This study was approved by VUMC Institutional Review Board (#150475). Written consent was obtained for use of genetic data (https://victr.vumc.org/biovu-consent/).

### Identifying adverse drug reactions in EHRs

For a given patient, allergy sections across all their clinical notes were extracted as free text. The data in an allergy section is often semi-structured (e.g., pcn [rash] and sulfa [itching]), but formatting can vary depending on the healthcare provider who entered the data. Therefore, drugs that appear in the allergy section were identified using case-insensitive regular expressions for generic names, brand names, abbreviations (e.g., pcn for penicillin), and common misspellings. Regular expressions allow us to match drug keywords within a drug allergy label irrespective of formatting. A full list of regular expressions used to identify drugs in this study can be found in S5 Table. The type of reaction was identified similarly with regular expressions when available. For drug allergy labels that refer to a class of drugs (e.g., penicillin, sulfa, etc.), we grouped all the drugs in the class as one ADR phenotype.

For each drug, we defined cases as individuals with any mention of the drug in the allergy section. For controls, we included individuals that met either of two criteria: 1) individuals who were prescribed the drug and had no mention of the drug in their allergy sections; or 2) individuals who only had labels for "no known drug allergy" or an equivalent description in of their allergy sections. For the first criteria, we used RxNorm codes–a normalized naming system for generic and branded drug–to identify individuals with prescriptions of the drug of interest.

### Genotyping and SNP imputation

Genotyping was performed on the Infinium Multi-Ethnic Genotyping Array (MEGAchip). We excluded DNA samples: (1) with per-individual call rate < 95%; (2) with wrongly assigned sex; (3) with a cryptic relationship closer than a third-degree relative (proportion identity by descent ≥0.25); or (4) unexpected duplication. We performed whole genome imputation using the Michigan Imputation Server (https://imputationserver.sph.umich.edu) [35] with the Haplotype Reference Consortium (HRC), version r1.1, [36] as reference. Principal components for ancestry (PCs) were calculated using common variants (MAF > 0.01) with high variant call rate (> 98%), excluding variants in linkage and regions known to affect PCs (HLA region on chromosome 6, inversion on chromosome 8 (8135000–12000000) and inversion on chr 17 (40900000–45000000), GRCh37 build). For association analyses, we used EasyQC (www.genepi-regensburg.de/easyqc) [37] to filter (1) poorly imputed variants with imputation quality ($R^2$) value of < 0.5, (2) EAF < 0.005, (3) deviation from Hardy-Weinberg equilibrium with a *P*-value $\leq 1\times10^{-6}$ and (4) variants with EAF that deviated from the HRC reference panel by > 0.3.

### Genetic analyses

All statistical analyses were performed with PLINK 2.0.[38] This study included 81,739 individuals from the Vanderbilt University Medical Center's BioVU DNA Biobank, [17] including GWAS data from 67,323 individuals with self-reported European ancestry and trans-ethnic validation using 14,416 individuals with self-reported African ancestry. We applied logistic regression models to investigate the association of genetic variants with risk of ADR to any of the 14 drugs or drug groups selected for this study. All regression models were adjusted for sex, age, length of EHR, and the first 10 principal components of the genotyping array for ancestry. Association results were annotated with ANNOVAR. [39] Region plots were

produced with LocusZoom. [40] Additional eQTL analysis used data from the Genotype-Tissue Expression (GTEx) project (www.gtexportal.org). [22]

For PheWAS, we used logistic regression models with phecodes, which are diagnosis codes aggregated into meaningful disease phenotypes that have been commonly used in phenome-wide analyses. [18,19] Patients with $\geq 2$ phecodes were assigned to case. All regression models were again adjusted for sex, age, length of EHR, and the first 10 principal components of the genotyping array for ancestry.

## Supporting information

**S1 Table. Summary of types of adverse drug reactions stratified by ancestry.**
(PDF)

**S2 Table. Genome-wide significant variants associated with adverse drug reactions in European ancestry individuals.**
(PDF)

**S3 Table. Trans-ethnic replication of lead variant per signal associated with adverse drug reactions in individuals with self-reported African ancestry.**
(PDF)

**S4 Table. Genome-wide significant variants associated with adverse drug reactions in self-reported African ancestry individuals.**
(PDF)

**S5 Table. Regular expressions for extracting adverse drug reactions from the 'Allergy section' of electronic health records.**
(PDF)

**S1 Fig. Manhattan plots of phenome-wide analysis of lead variants in *CYP2D6*, *OPRM1*, *PTHLH*, and *HLA/MICA* associated with adverse drug reactions (ADRs).** Red lines on Manhattan plots show the phenome-wide level of significance ($5.0 \times 10^{-5}$). Phenotypes with P-values < 0.005 were annotated.
(PDF)

## Author Contributions

**Conceptualization:** Neil S. Zheng, Elizabeth J. Phillips, Wei-Qi Wei.

**Formal analysis:** Neil S. Zheng.

**Funding acquisition:** C. Michael Stein, Dan M. Roden, Joshua C. Denny, Elizabeth J. Phillips, Wei-Qi Wei.

**Investigation:** Neil S. Zheng.

**Methodology:** Neil S. Zheng, Wei-Qi Wei.

**Resources:** Wei-Qi Wei.

**Software:** Neil S. Zheng, Lan Jiang, Christian M. Shaffer, QiPing Feng.

**Writing – original draft:** Neil S. Zheng.

**Writing – review & editing:** Cosby A. Stone, V. Eric Kerchberger, Cecilia P. Chung, QiPing Feng, Nancy J. Cox, C. Michael Stein, Dan M. Roden, Joshua C. Denny, Elizabeth J. Phillips, Wei-Qi Wei.

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
