## [Decision Letter · Decision Letter 0]

1 Mar 2021

Dear Dr Wei,

Thank you very much for submitting your Research Article entitled 'High-throughput genetic analyses of adverse drug reactions using electronic health records' to PLOS Genetics.

The manuscript was fully evaluated at the editorial level and by independent peer reviewers. The reviewers appreciated the attention to an important problem, but raised some substantial concerns about the current manuscript. Based on the reviews, we will not be able to accept this version of the manuscript, but we would be willing to review a much-revised version. We cannot, of course, promise publication at that time.

If you decide to revise the manuscript for further consideration at PLOS Genetics, please aim to resubmit within the next 60 days, unless it will take extra time to address the concerns of the reviewers, in which case we would appreciate an expected resubmission date by email to plosgenetics@plos.org.

[LINK]

We are sorry that we cannot be more positive about your manuscript at this stage. Please do not hesitate to contact us if you have any concerns or questions.

Yours sincerely,

Gregory M. Cooper, PhD

Associate Editor

PLOS Genetics

Scott Williams

Section Editor: Natural Variation

PLOS Genetics

Reviewer's Responses to Questions

**Comments to the Authors:**

Reviewer #1: While I agree that the drug allergy section in the EHR often lists ADRs, true allergies are also reported and captured (PMID:30100688 Figure 2). I am concerned that combining anything listed in this section for a certain drug does not provide an appropriate phenotype. Given that allergy data is not uniformly collected and/or reported and can represent everything from mild, expected, patient-reported side effects to hypersensitivity reactions, knowing the types of “allergies” reported is important to be able to determine risk-benefit. Grouping these effects in this manner assumes that genetic factors associated with gastrointestinal patient-reported side effects from antibiotics are the same that contribute to the development of Stevens Johnson Syndrome or other hypersensitivity reactions. While the authors address the inability to ascertain the specific drug reaction as a limitation, they also state that in their previous study (Ariosto 2014), the majority of patients had gastrointestinal effects when opioid allergy alerts were evaluated. Therefore, it seems that the type of allergy/ADR can be obtained in some circumstances. Would like to see a breakdown of the types of drug reactions that were included for each drug/drug class when this data was available. Without this information, it is difficult to determine (1) if the phenotype was appropriate for the GWAS and (2) what the results actually indicate. Broadly indicating a genetic variant is associated with an ADR does not provide any information on the potential risk of the medication for the patient. Expected GI side effects and hypersensitivity reactions would not result in the same clinical management decisions, and deeming a patient "at risk" for an ADR/allergy could cause the patient to be given suboptimal therapy when the risk was actually only a common, expected side effect. Can the authors comment on this, as this was not addressed in the manuscript, and is a major limitation.

Page 12: The authors conclude that this framework provides insights for optimizing drug therapy with maximal efficacy and minimal adverse effects. While this approach may help minimize adverse effects, it does not provide information on treatment efficacy, as lack of an ADR/allergy does not indicate that the drug is effective. Recommend rewording.

Page 12: The authors state that “utilizing EHRs allowed us to study ADRs in individuals with diverse clinical and ethnic backgrounds.” However, the GWAS was not conducted in patients of African ancestry and the identified associations were not validated in these patients, so this did not seem to be the best the method to study ADRs in non-Europeans. I am wondering (1) If the top 10 most reported drugs in the allergy section were the same for patients of European and African ancestry, and (2) given that there are known ancestral differences in susceptibility to certain ADRs (e.g. ACE inhibitor-induced angioedema in African ancestry), other than sample size, was there a reason the GWAS not also conducted in the African ancestry individuals?

Given that the associations did not validate in African Americans, was any attempt made to validate the identified associations in individuals of European ancestry? Since previously identified and replicated associations for statin-induced myopathy (i.e. SLCO1B1) and aspirin-associated ADRs, such as aspirin-induced asthma (HLA-DPB1), were not identified in the study, I am wondering if this could be due to how the drug allergy phenotype was defined, or if the authors have another explanation.

Reviewer #2: Zheng et al report the results of a genome-wide association study of adverse drug reactions using the “allergy section” of the electronic health-care records of a large medical center coupled to a biobank. The authors investigated 14 common drug/drug groups and report 7 genetic loci associated with ADRs at genome wide significance of p< 5 x 10-8. A considerable strength of the manuscript is the large sample size for a PGx analysis including ~81.7k participants combined with the use of EHR data. The authors may consider the following points to improve their manuscript.

Major comments:

• While a strength of the paper is the considerable size, the reported findings are confirmatory and lack novelty unfortunately. In fact the paper is more a method validation than a discovery study and this should be reflected in the title and throughout the paper. Therefore I would encourage to show more data on the text mining of the “allergy section”

• It is not clear to me how the 14 drugs/groups were selected. This should be described in more detail. From the title I would have thought that the authors for instance would have focused on severe hypersensitivity reactions i.e. stevens–Johnson syndrome, toxic epidermal necrolysis, drug-induced liver injury.

• In the introduction the authors correctly address the challenge that “Drug response phenotypes, such as ADRs, are less often recorded than physiological traits and common diseases” To be able to assess drug response phenotypes detailed information regarding dose, interval and time relation with occurrence of the ADR is important. Was more detailed information regarding dose and time of administration available from the EHR and considered for inclusion in the analysis?

• The authors correct for multiple testing of the genetic variant but it is not clear how many endpoints were tested and if this is corrected for

• The methods section is quite brief and should be expanded. For example, it would be useful to include detailed information on the identification of ADRs from the allergy sections free text.

Minor comments

• The paper is very brief and little detail is provided. For example, the results section on the PheWAS is only 2 sentences.

• The figures are of low resolution and should be improved

**Have all data underlying the figures and results presented in the manuscript been provided?**

Reviewer #1: None

Reviewer #2: **No: **Authors claim data are part of EHR and cannot be shared

PLOS authors have the option to publish the peer review history of their article (what does this mean?). If published, this will include your full peer review and any attached files.

Reviewer #1: No

Reviewer #2: No

---

## [Editor Report · Decision Letter 1]

10 May 2021

Dear Dr Wei,

We are pleased to inform you that your manuscript entitled "High-throughput framework for genetic analyses of adverse drug reactions using electronic health records" has been editorially accepted for publication in PLOS Genetics. Congratulations!

Yours sincerely,

Gregory M. Cooper, PhD

Associate Editor

PLOS Genetics

Scott Williams

Section Editor: Natural Variation

PLOS Genetics

Comments from the reviewers (if applicable):

**Data Deposition**

http://datadryad.org/submit?journalID=pgenetics&manu=PGENETICS-D-20-01665R1

**Press Queries**

---

## [Editor Report · Acceptance letter]

25 May 2021

PGENETICS-D-20-01665R1 

High-throughput framework for genetic analyses of adverse drug reactions using electronic health records 

Dear Dr Wei, 

We are pleased to inform you that your manuscript entitled "High-throughput framework for genetic analyses of adverse drug reactions using electronic health records" has been formally accepted for publication in PLOS Genetics! Your manuscript is now with our production department and you will be notified of the publication date in due course.

With kind regards,

Zsofi Zombor

PLOS Genetics

On behalf of:
